# Antimicrobial Activity and Cytotoxicity of Nonsteroidal Anti-Inflammatory Drugs against Endodontic Biofilms

**DOI:** 10.3390/antibiotics12030450

**Published:** 2023-02-23

**Authors:** Carmen María Ferrer-Luque, Carmen Solana, Beatriz Aguado, Matilde Ruiz-Linares

**Affiliations:** 1Department of Stomatology, University of Granada, 18071 Granada, Spain; 2Instituto de Investigación Biosanitaria ibs.GRANADA, 18012 Granada, Spain; 3Independent Researcher, 18194 Granada, Spain

**Keywords:** antimicrobial activity, endodontic biofilms, diclofenac, ibuprofen, non-steroidal anti-inflammatory drugs

## Abstract

Persistent infections have become a challenge in dentistry because of growing antibiotic resistance. Nonsteroidal anti-inflammatory drugs (NSAIDs) appear to be a therapeutic alternative to control biofilm infection. The objective of this work is to evaluate the antimicrobial activity and cytotoxicity of sodium diclofenac (DCS), ibuprofen (IBP) and ibuprofen arginine (IBP-arginine) solutions against endodontic polymicrobial biofilms. Sterile radicular dentin blocks of 4 mm × 4 mm × 0.7 mm were used as substrate to grow biofilm. The dentin blocks were submerged into solutions for 5 min. The antimicrobial activity was evaluated by means of the adenosine triphosphate (ATP) assay and confocal laser scanning microscopy (CLSM). Fibroblasts 3T3-L1 (ECACC 86052701) were used to test the cytotoxicity of irrigating solutions. The antibiofilm effects determined by the ATP assay showed that 4% IBP-arginine solution exerted the highest antibiofilm activity, followed by 4% DCS and 4% IBP, with statistical differences among groups (*p* < 0.001). As for CLSM, 4% DCS and 4% IBP-arginine solutions gave the lowest viable cell percentages, without significant differences between them. Cytotoxicity results at 1/10 dilution were similar for all solutions. At 1/100 dilution, a 4% DCS solution obtained the lowest cell viability for both time periods assayed, 1 h and 24 h. The IBP-arginine group showed the highest cell viability at 24 h. In this preliminary study, in terms of antibiofilm activity and cytotoxicity, a mixed 4% IBP-arginine solution gave the most promising results. NSAID solutions could be recommendable drugs for endodontic disinfection procedures.

## 1. Introduction

Apical periodontitis is an inflammatory disease caused mainly by bacteria organized as microbial biofilms attached to the dentin of the root canal system. Infection progresses apically to reach and establish itself in periradicular tissues, where the microorganisms promote the immune response from the host and determine symptomatic or asymptomatic disease nature and periradicular tissue damage [1,2,3].

Persistent infections, usually associated with biofilm-producing bacteria, are a challenge in dentistry today due to the emergence of antibiotic resistance. The World Health Organization recognizes that antimicrobial resistance is one of the greatest threats to global health, and antibiotic resistance is of particular concern in dentistry as they are the drugs most prescribed by professionals (10% of antibiotic prescriptions are issued by dentists) [4]. Endodontic infections generally result from untreated deep caries. When root canal infection is not treated, it can invade the periapical tissues and lead to an inflammatory disease, which sometimes produces abscesses and can even cause systemic health issues that require hospitalization. In such cases, antibiotics are needed to resolve the infectious process. Yet their indiscriminate use in oral disease treatment has generated a significant resistance to these substances [5,6]. In addition, the search for novel antibacterial agents for use in endodontic therapy is fundamentally determined by the resistance of microorganisms to antimicrobial substances. In this sense, great efforts are being made toward drug repurposing for biofilm control in infectious processes, where antibiotic performance is diminished due to the resistance generated by their indiscriminate use.

One possible therapeutic alternative would be nonsteroidal anti-inflammatory drugs (NSAIDs), medication commonly used to relieve pain and inflammation [7], also demon-strating antimicrobial activity [8] and seen as a promising strategy to control infection [9,10]. Their use as antimicrobial drugs could be recommended as a complementary or alternative therapy for infections related to biofilm [9,11].

In endodontic therapy, root canal disinfection is the first step to resolve apical periodontitis [12]. To this end, irrigating solutions [13] and intracanal medication [14] may control infection [15] and restore health to affected tissues. Ibuprofen (IBP) is a widely used NSAID that pertains to aryl-propionic acid derivatives. It acts mainly on cyclooxygenase enzymes (COX-1 and COX-2), inhibiting the synthesis of their metabolites, such as PGE2 [7]. Its free carboxyl group allows for structural modification and is also responsible for side effects that could appear at a gastric, renal or hepatic level [16]. However, these effects are not relevant in light of the possible antimicrobial action of NSAID drugs in endodontic therapy. In root canal treatment, IBP used as intracanal medication showed greater antibacterial activity in vitro [17] against *Enterococcus faecalis* than Ca(OH)_2_ paste; and in vivo, similar effectiveness in teeth with asymptomatic apical periodontitis [18]. 

Sodium diclofenac (DCS) is another NSAID that belongs to the phenylacetic acid family and binds cyclooxygenase (COX)-2 enzyme with greater potency than COX-1 [19] to inhibit the synthesis of prostanoids, prostacyclin and thromboxane [20]. It has proven bactericidal action against different bacteria [8,9,17] and antibiofilm effects when applied in solution form [21] and is associated with calcium hydroxide paste [22,23] against *E. faecalis* biofilm, as well as against polymicrobial endodontic biofilms when mixed with tricalcium silicate-based cements [24] and in newly formulated hydrogels [25].

When NSAIDs are associated with antibacterial drugs, the antimicrobial activity is usually enhanced [26]. In this sense, a formulation of IBP could include in its composition arginine, a prebiotic disruptor that affects the dynamic microbial interactions related to biofilm development [27]. L-arginine is a natural amino acid found in micromolar concentrations in saliva and plaque biofilm. It is one of the few nutrients used as a strategy to target pathogens without destroying the physiologically present microbial flora. It can be metabolized by the arginine deiminase system of certain bacteria, which helps raise intraoral pH [28]. Moreover, it modulates pathogen growth [29], EPS matrix production [27] and biofilm biomass, to maintain homeostasis [30].

The antibiofilm effects of arginine on mono-species [31] and multi-species cariogenic biofilm [32] are related to polysaccharide reduction and the prebiotic concentration used. To date, however, its antimicrobial activity against polymicrobial biofilms obtained from necrotic root canals of teeth with apical periodontitis is unknown. Considering that irrigating solutions in root canal treatment may come in direct contact with periradicular tissues, it is also fundamental to evaluate the biocompatibility of these drugs. Accordingly, the aim of this work is to evaluate the antimicrobial activity and cytotoxicity of DCS, IBP and IBP-arginine solutions against polymicrobial endodontic biofilms.

## 2. Results

### 2.1. Antimicrobial Effects

The antimicrobial effects of solutions as determined by the ATP assay showed that 4% IBP-arginine exerted the highest antibiofilm activity, followed by 4% DCS and 4% IBP, with statistically significant differences among groups (*p* < 0.001). For CLSM, the Log_10_ total biovolume in NSAIDs groups was similar and showed a scarce but significant reduction with respect to the control. The 4% DCS and 4% IBP-arginine groups gave the lowest viable cell percentages, without significant differences between them. In contrast, the 4% IBP group showed the highest percentage of viable cells, with statistically significant differences from the other groups. Data on antimicrobial effects of the solutions are indicated in Table 1. Representative scanning electron microscope images of polymicrobial endodontic biofilm growth in dentin specimens are displayed in Figure 1. Representative images of the treated biofilms can be found in Figure 2.

### 2.2. Cytotoxicity

Results at 1/10 dilution were similar for all solutions, regardless of the time evaluated (1 h, *p* = 0.758; 24 h, *p* = 0.943) without significant differences at 1 and 24 h. At 1/100 dilution there were statistically significant differences between solutions at the two time periods (*p* < 0.001). After 1 h of exposure, the 4% DCS (41.93%) obtained the lowest cell viability, followed by IBP-arginine (62.14%) and IBP (78.89%), with significant differences between them. At 24 h, 4% DCS continued to show the highest cytotoxicity, while both groups of IBP obtained substantial viability values, with no significant differences between them. Comparisons between 1 and 24 h gave no significant differences for 4 % IBP-arginine, but there were differences regarding 4% DCS and 4% IBP. Data on the viability percentage of 3T3-L1 fibroblasts as determined by optical density after contact with the solutions are offered in Table 2.

## 3. Discussion

An increasing number of publications suggest that NSAIDs exert antimicrobial and anti-biofilm activities against clinically relevant bacteria [8,9,17,18,21,22,23,24,25,33]. Since these drugs are universally used, such indications of eliminating biofilms and improving disinfection in root canals are promising. 

In this study, all NSAIDs tested showed antimicrobial efficacy against a polymicrobial endodontic biofilm. The most interesting result was for the 4% IBP-arginine solution, with a 96% reduction of RLUs, followed by 4% DCS (88%) and 4% IBP alone (78%). The incorporation of arginine into ibuprofen apparently provides for a significant increase in its antimicrobial activity. Arginine is absorbed and metabolized by biofilm cells, and alkali generation—via arginine catabolic pathways—improves pH homeostasis [30]. This is an important factor in apical periodontitis, given the presence of local acidosis in the root canal and the surrounding tissues [34].

In turn, L-arginine contains guanidine [35], a molecule that is protonated with a positive charge and interacts electrically with negatively charged bacterial cell surfaces to produce bacteriostatic effects. Guanidine-containing compounds have shown efficacy against Gram-positive and Gram-negative bacterial strains, as well as modulating effects on drug resistance [35]. L-arginine may act like a surface-active agent when interacting with bacterial membranes, increasing the drug’s availability inside the cells, and thus enhancing its antimicrobial effectiveness [36]. 

The 4% DCS solution (pH 7.8) gave intermediate values in antimicrobial activity. The antimicrobial mechanism action of DCS may be related to an inhibition of microbial cell wall synthesis, altered membrane function or membrane damage, and the inhibition of nucleic acid synthesis [8,9,10]. It was recently shown that DCS solutions at 5% and 2.5% have greater antimicrobial effects than triple antibiotic and double antibiotic solutions at 1 mg/mL on *E. faecalis* biofilm [21] and can be considered a valid alternative for controlling the infection of teeth with apical periodontitis. The addition of 5% and 10% DCS to a hydraulic calcium silicate-based cement [24] enhanced disinfection in a concentration- and time-dependent manner. In this context, DCS hydrogel formulations demonstrate higher antimicrobial efficacy than antibiotic gels or a calcium hydroxide paste on root canal polymicrobial biofilms [25]. 

The CLSM results appear to support those obtained with the ATP assay, with the lowest percentage of viable cells for 4% DCS and 4% IBP-arginine, without significant differences among them, but giving statistically significant differences with respect to a 4% IBP solution. In this way, the viability results of both tests can be considered complementary, and discrepancies may be attributed to differences in the methodologies used. The ATP determination requires the disruption of the biofilms adhered to dentin for cell recovery, while CLSM evaluates the percentage of viable cells in situ, in a randomized number of dentin areas.

Although the objective of this work was to evaluate the antimicrobial activity and cytotoxicity of ibuprofen and diclofenac solutions against an endodontic biofilm, we considered it appropriate to include a solution of ibuprofen combined with arginine given its efficacy against different oral biofilms [31,32], as well as the proven synergistic effects when combined with antibiotics in antimicrobial complexes [36,37]. Still, in view of the results obtained here, further investigations are needed, including arginine solutions alone and at different concentrations, to assess their true potential against polymicrobial root canal biofilms. 

In addition to meeting antimicrobial requirements, an ideal irrigating solution must ensure biocompatibility [38]. Two dilutions were chosen to test the cytotoxicity effects of experimental groups, given that the amount and concentration at which the solution reaches the apical region is uncertain [38]. There were significant differences between the tested dilutions, the greatest cytotoxicity being seen for the 1/10 dilution. At this dilution, all solutions showed similar cell viability values at 1 and 24 h, and there were no significant differences among them.

The values of cytotoxicity obtained by ibuprofen solutions at dilution 1/100 showed the highest viability percentages for the two times tested. With 4% IBP-arginine, similar cell viability was obtained at 1 h and 24 h (62.14% and 60.18%, respectively). Moreover, the 4% IBP-arginine group was the least cytotoxic at 24 h, showing no statistical differences from 4% IBP, which might indicate that arginine lends stability to the solution [36]. The viability percentage of 4% DCS was lower than that observed for the two IBP solutions.

Its known that the systemic administration of DCS has a negative effect on bone healing in animals [39]. As local medication, however, 5% IBP and 5% DCS associated with Ca(OH)_2_ pastes have shown low cytotoxicity in a mouse pre-osteoblast cell line [23]. Similarly, a tricalcium silicate-based experimental cement containing sodium diclofenac exerted the same cytotoxic effects as MTA [40]. The alkaline environment (pH > 10) most likely contributed to the low toxicity found [22]. Although the current study is the first in vitro investigation analyzing the cytotoxicity of these drugs in solution forms, the results cannot necessarily be extrapolated to the clinical situation, which is acknowledged as a limitation. More studies are needed to corroborate the results obtained, including ex vivo and in vivo research on these solutions and newly improved formulations.

The inclusion of NSAIDs constitutes a therapeutic alternative in endodontic treatments. In clinical protocols, they could be used as final irrigating solutions to enhance the disinfection achieved after interventions, and additionally might reduce postoperative pain. In addition, their use in regenerative endodontic procedures instead of antibiotics would contribute to reducing our resistance to these drugs. Furthermore, they could serve as an alternative drug for patients with an allergy to sodium hypochlorite, the reference antimicrobial of routine use in endodontics. Still, despite the promising results of NSAID solutions in terms of antibiofilm effects and biocompatibility, knowledge about other desirable properties is needed, such as remnant pulp tissue dissolution or smear layer and debris removal. Future studies should be carried out on microbiological samples collected from failed root-filled teeth to evaluate treatments closer to clinical reality.

Currently, the antibiotic resistance approach focuses on repositioning drugs that already exhibit satisfactory results within other medical fields [6]. In endodontics, the persistent nature of microbiota in root canals after their treatment calls for further efforts in disinfection strategies [41]. In this case, in terms of antibiofilm activity and cytotoxicity, a mixed 4% IBP-arginine solution showed the most promising results.

In conclusion, NSAIDs could be suitable drugs for the control of endodontic infection. In this work, ibuprofen mixed with arginine was the least cytotoxic and most effective antimicrobial solution against a polymicrobial root canal biofilm. Diclofenac sodium was more cytotoxic than both ibuprofen solutions and showed less antibiofilm efficacy than the IBP-arginine. The results obtained here should be confirmed in future investigations when the tested solutions might be recommended for clinical use.

## 4. Materials and Methods

The study protocol (no. 1076 CEIH/2020) was approved by the Research Ethics Committee of the University of Granada, Spain; informed consent was obtained from all patients before collection of the microbiological samples or extracted teeth. A PRILE 2021 flowchart (Figure 3) summarizes the key steps of this laboratory study [42].

### 4.1. Antimicrobial Activity Determination

#### 4.1.1. Preparation and Infection of Dentin Specimens

The sample size was estimated based on studies comparing the efficacy of antimicrobial agents against microbial biofilms grown in dentin [21,22,25]. Accordingly, for the antimicrobial tests, 20 (*n* = 20) dentin specimens were allocated for ATP; 5 dentin specimens per group for CLSM with 4 measurements/specimen (*n* = 20 stacks/group); and 4 specimens as sterility controls and 8 for biofilm growth confirmation by SEM.

For the preparation and infection of dentin specimens, sterile radicular dentin blocks (*n* = 112) served as the biofilm substrate, following a previously reported protocol [21]. Briefly, 56 freshly extracted noncarious single-rooted human teeth with closed apexes, and no cracks or resorptive defects, were selected. After removal of debris, calculus and soft tissues on the root surface, the teeth were stored in 0.1% thymol solution at 4 °C until use. Selected dentin specimens were obtained, discarding the crowns and the middle and apical thirds of the roots. The coronal portion of the root was divided longitudinally into two halves, and the outer cementum of each half was removed. The inner part of the dentin root was progressively polished with 220- to 800-grit silicon carbide papers to create a flat surface. After adjusting the size using a caliper to obtain 4 mm × 4 mm × 0.7 mm (width × length × height) specimens, the smear layer that had formed during the preparation of the specimens was removed with 17% EDTA for 5 min. All samples were then rinsed with distilled water for 10 min and sterilized in an autoclave at 121 °C for 20 min. To check the sterility of the dentin, each specimen was incubated in 5 mL BHI at 37 °C for 24 h, whereby the absence of turbidity in the culture medium could be verified.

The microbial samples studied here were collected from root canals of single-rooted necrotic teeth of three volunteers with apical periodontitis, by means of a strictly aseptic technique [25]. Having filled the root canal with sterile saline solution, taking care to avoid overflow, a sterile #15 K file (Dentsply Sirona, Ballaigues, Switzerland) was introduced 1 mm short of the apical foramen, and gentle filing motion was undertaken for 30 s before removal. Then, three sterile paper points were inserted into the root canal, and left inside for 1 min to absorb the fluid. Both the files and the paper points were put into Eppendorf tubes with Tris-EDTA buffer and frozen at −20 °C until use.

To appraise the dentin infection, the microbiological samples were mixed in 5 mL of brain–heart infusion broth (Scharlau Chemie SA, Barcelona, Spain) supplemented with 5 g/L yeast extract and 5% (*v*/*v*) vitamin K and hemin (BHI-YE). After anaerobic incubation at 37 °C for 48 h, the cell density was adjusted in a spectrophotometer—to a concentration of approximately 3.0 × 10^7^ colony-forming units per mL in BHI-YE broth [25]. 

The wells of 24-well microtiter plates were inoculated with 200 μL of the microbial suspension plus 1.8 mL sterile BHI. After submerging the sterile dentin blocks in the inoculated wells, they were incubated under anaerobic conditions for 3 weeks at 37 °C. The BHI was refreshed once a week. As the sterility control, four dentin blocks were inoculated with sterile BHI throughout the experimental procedure. Eight specimens were processed for observation under scanning electron microscopy (SEM) to confirm the growth of biofilms. To this end, the dentin specimens were fixed in 2.5% glutaraldehyde in 0.1 M cacodylate buffer, pH = 7.4, for 2 h, at 4 °C. Afterwards, they were rinsed in the same buffer (3 times, 15 min each at 4 °C) and serially dehydrated by a gradient of increasing ethanol concentrations (50%, 70%, 90% and 100%, for 15 min with each concentration). They were then dried using a critical point dryer (Polaron CPD7501), sputter-coated with gold (Polaron E5000), and finally examined with SEM (TESCAN AMBER X, GmbH, Dortmund, Germany) under high resolution conditions using a detector in the column of secondary electrons, operating at 5 Kev. 

#### 4.1.2. Antimicrobial Test

The antimicrobial activity was evaluated by means of the adenosine triphosphate (ATP) assay (BacTiter-Glo; Promega, Madison, WI, USA) and confocal laser scanning microscopy (CLSM).

Eighty dentin specimens were used for ATP determination. They were washed with saline solution for 1 min and randomly divided into 4 groups (*n* =20) according to the drug used (Table 1). All the test solutions were of Spanish Pharmacopeia grade and master formulation: group 1, 4% DCS (pH 7.8); group 2, 4% IBP (pH 4.03); group 3, 4% IBP-arginine (pH 8.12) and group 4, 0.9% saline solution (SS) as the control group. The concentrations of the tested solutions were decided based on previous results of diclofenac sodium against endodontic biofilms [21,22,25]. The solutions were prepared by dissolving 4 g of DCS, IBP and IBP-arginine (1:1) in 100 mL of distilled water. Methylcellulose and Cremophor^®^ RH 40 were used as excipients.

The dentin blocks were submerged in 120 μL of the solutions for 5 min. Then the specimens were placed in Eppendorf tubes with 200 μL BHI, stirred in a vortex for 10 s, and sonicated for 10 min to ensure the recovery of biofilms. For bacterial count determination, 100 μL of suspension was added to 100 μL BacTiter-Glo reagent in a 96-well white plate (Greiner, Monroe, NC, USA), followed by incubation at room temperature for 5 min. The luminescence produced was measured with a luminometer (GloMax; Promega, Madison, WI, USA), expressed as absolute values of relative light units (RLUs) in Table 1. The reduction percentage RLUs with respect to the control was calculated using the following formula: (1 − [RLUs test/mean RLUs control]) × 100.

For the purpose of CLSM evaluation, 20 infected dentin specimens were randomly divided into four groups (*n*= 5/group) according to the solutions specified in Table 1. Specimens were washed with saline solution for 1 min and then submerged in the antimicrobial solutions for 5 min. After this contact period, the samples were rinsed again with 0.9% saline solution and labeled with the LIVE/DEAD^®^ kit (BacLight™; Invitrogen, Eugene, OR, USA) which allows for the evaluation of the integrity of the cell membrane [43]. It contains two fluorescent components, A and B, consisting of Syto-9 and Propidium Iodide (PI), respectively. Syto-9 is a green fluorescent dye, which stains nucleic acids. Propidium iodide (IP) is a red fluorescent dye that only stains cells with damaged cytoplasmic membranes (considered as dead microbes). After staining the samples with a 1:1 mixture of SYTO-9 and IP for 15 min, at room temperature and protected from light, they were washed with saline solution and mounted on a 60 l-Dish (Ibidi, Martinsried, Germany) with mounting oil (BacLight™, Invitrogen, Madrid, Spain) for direct observation using an inverted confocal laser microscope (Leica TCS-SP5 II; Leica Microsystems, Mannheim, Germany). An independent and experimented examiner, blind to the study group, performed the scanning. The absorption and emission wavelengths were, respectively, 494/518 nm for Syto 9, and 536/617 nm for PI. Four microscopic confocal volumes (stacks) from equidistant random areas were acquired from each specimen (*n* = 20 stacks/group) —from the top of the biofilm to the dentin surface, using the 40× oil lens, 1 μm step size, and a format of 512 × 512 pixels. Each picture represented an area of 387 μm × 387 μm. 

For quantification, *bio*Image_L software (www.bioImageL.com (accessed on 11 January 2023)) was used [44]. Evaluated in each group were the variables: Log_10_ total biovolume and the percentage of cells with intact membranes (green stained population, considered as live cells). They were calculated as green population/(green population + red population).

Statistical analysis relied on SPSS 23.0 software (SPSS Inc., Chicago, IL, USA). Data on the green percentage were subjected a priori to the Anscombe transformation. The Kolmogorov–Smirnov test showed a non-normal distribution for all variables. A global comparison was performed by means of the Kruskal–Wallis test followed by the Mann–Whitney U test for pair-by-pair comparison.

### 4.2. Cytotoxicity Assessment

The cytotoxicity of irrigating solutions was tested on 3T3-L1 fibroblasts (ECACC 86052701). Cells were grown in a 75-cm^2^ culture flask in Dulbecco’s modified Eagle medium (DMEM; Gibco, Thermo Fisher Scientific, Paisley, UK) supplemented with 10% inactivated fetal bovine serum (FBS; Gibco), 2 mM glutamine, and antibiotics (100 U/mL penicillin and 100 Pg/mL streptomycin; Gibco). Cells were maintained at 37 °C in a humidified atmosphere with 5% CO_2_. Confluent cells were detached using EDTA solution (0.5 mM EDTA pH 8.0 in PBS), the supernatant was centrifuged (1000 rpm for 10 min) and the pellet was resuspended in DMEM containing 10% FBS. The cells were subsequently counted in a Neubauer chamber (Brand GmbH + CO KG, Wertheim, Germany). Adherent cells in a logarithmic growth phase were seeded in 96-well flat-bottom microtiter plates (Jet Biofil, Guangzhou, China) at a concentration of 10^4^ cells/well, then incubated at 37 °C for 24 h with 5% CO_2_ [45]. 

The solutions were tested at dilutions of 1/10 and 1/100. After 1 h and 24 h of exposure (100 μL/well), all solutions were removed and the cells were incubated with 10 μL of CellTiter 96^®^ AQueous One Solution Cell Proliferation Assay (MTS), as recommended by the supplier (Promega, Madison, WI, USA). This assay was performed in quadruplicate and repeated three times. The optical density (OD) was measured at 490 nm using a spectrophotometer (FLUOstar Optima, Ortenberg, Germany). The values of OD were expressed as the percentage of cell viability with respect to the control groups (cells in DMEM) using the following formula:Viability (%) = Mean OD (test)/Mean OD (control) × 100

The data on cell viability percentage, having been subjected to the Anscombe transformation, showed a normal distribution according to the Kolmogorov–Smirnov test. A global comparison was undertaken, performed by ANOVA and the Tukey post hoc test, to derive groupings. Comparisons between 1 and 24 h for each dilution, as well as the two dilutions at each exposure time, were in turn determined by means of the Student’s *t* test.

## Figures and Tables

**Figure 1 antibiotics-12-00450-f001:**
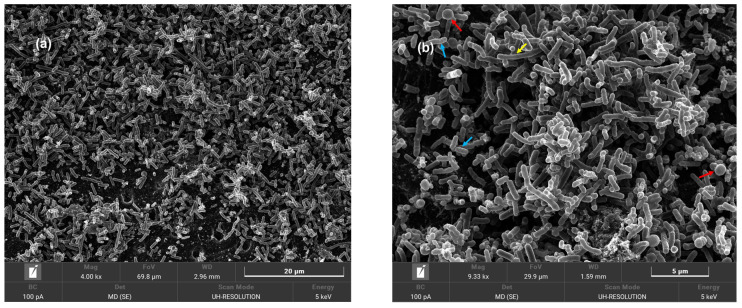
Representative scanning electron microscopy images showing the growth of polymicrobial biofilms in infected dentin samples: (**a**) ×4000; (**b**) ×13,000. Note the diversity of microbial composition (different colored arrows represent different species and bacterial shape types) grown within close proximity. Antagonistic and cooperative synergistic microbial interactions play an essential role in ecological regulation and endodontic biofilm formation.

**Figure 2 antibiotics-12-00450-f002:**
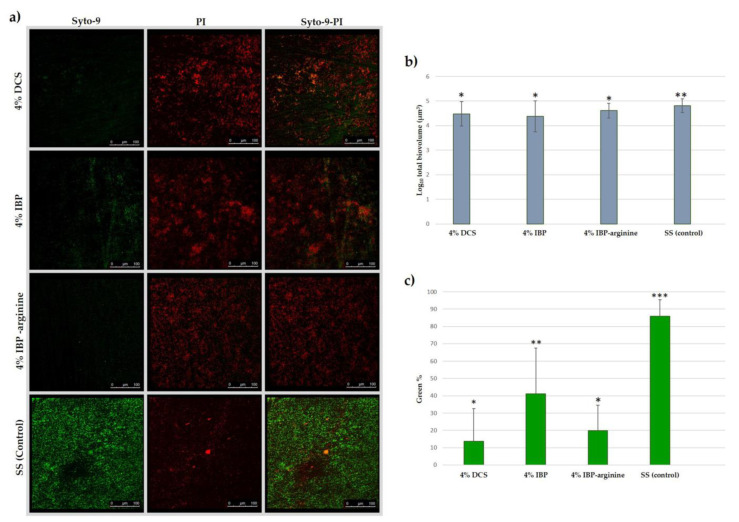
(**a**) Representative images of 3-week biofilms treated with diclofenac (DCS), ibuprofen (IBP), IBP-arginine and saline solution (SS) visualized under CLSM after staining with Syto-9/PI. Syto-9 stained nucleic acid and emitted green fluorescence (considered as live cells), whereas damaged cells were stained by PI (red fluorescence for dead bacteria); (**b**) the Log_10_ total biovolume in the groups was similar and showed a scarce but significant reduction with respect to the control. (**c**) Mean (SD) percentage of green (live) cells by group (*n* = 20 stacks/group). * The 4% DCS and 4% IBP-arginine groups gave the lowest viable cell percentages, without significant differences between them (*p* < 0.001). ** The 4% IBP group showed the highest percentage of viable cells, having statistically significant differences from the other groups. *** The control group showed statistically significant differences with all the experimental groups.

**Figure 3 antibiotics-12-00450-f003:**
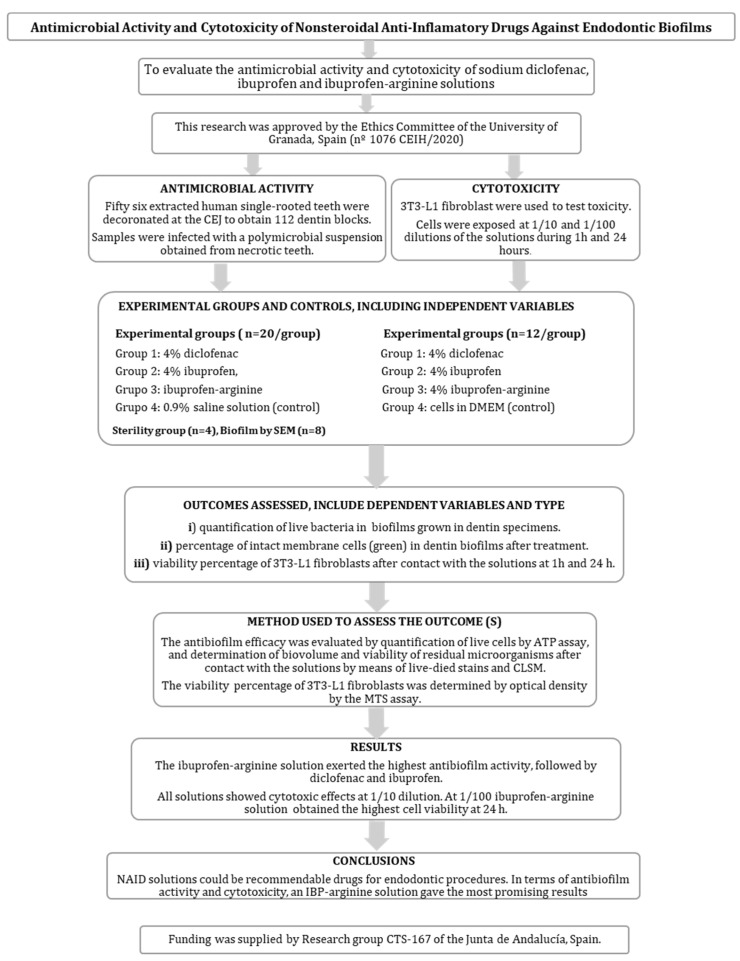
PRILE 2021 flowchart.

**Table 1 antibiotics-12-00450-t001:** Antimicrobial activity of solutions on polymicrobial biofilms. Relative light units (RLUs), Log_10_ biovolume (μm^3^), and green percentage. Mean ± standard deviation (SD).

Solutions	Relative Light Units *n* = 20 Samples/Group	Biovolume Log_10_ *n* = 20 Stack/Group	Green Percentage
4% DCS	42,519.35 (35,611.49) ^a^	4.48 (0.50) ^a^	13.66 (19.01) ^a^
4% IBP	78,835.60 (35,920.70) ^b^	4.37 (0.63) ^a^	41.19 (26.33) ^b^
4% IBP-arginine	12,053.65 (3482.41) ^c^	4.61 (0.30) ^a^	19.84 (14.68) ^a^
0.9% SS (control)	361,297.90 (189,230.83) ^d^	4.81 (0.28) ^b^	85.91 (9.60) ^c^
Comparison *p* value *	<0.001	0.031	<0.001

Data of green percentage were previously subjected to the Anscombe transformation. * Global comparison was performed using the Kruskal–Wallis test followed by the Mann–Whitney U test for pair-by-pair comparison. Read vertically, the same lower-case letters (a–d) show differences that are not statistically significant. DCS, sodium diclofenac; IBP, ibuprofen; IBP-arginine, ibuprofen arginine; SS, saline solution.

**Table 2 antibiotics-12-00450-t002:** Viability percentage of 3T3-L1 fibroblasts determined by optical density after contact with the solutions at two time periods and dilutions. Mean ± standard deviation (SD). *n* = 12/group.

Time	Dilutions	Diclofenac	Ibuprofen	Ibuprofen Arginine	*p*-Value *
1 h	1/10	35.14 ± 11.73 ^a^	38.35 ± 11.35 ^a^	37.95 ± 11.83 ^a^	0.758
	1/100	41.93 ± 11.48 ^1,A^	78.89 ± 12.44 ^2,A^	62.14 ± 11.71 ^3,A^	<0.001
	*p*-value **	0.163	<0.001	<0.001	
24 h	1/10	29.95 ± 9.1 ^a^	31.25 ± 9.59 ^a^	30.38 ± 9.28 ^a^	0.943
	1/100	30.1 ± 8.97 ^1,B^	50.31 ± 12.283 ^2,B^	60.18 ± 9.59 ^2,A^	<0.001
	*p*-value **	0.963	<0.001	<0.001	

Data on controls (cells in DMEM); mean (SD) at 1 h: 1.11 (0.16) and 24 h: 1.36 (0.11). Global comparison by ANOVA * and post hoc Tukey test. Pair-by-pair comparison by Student’s *t* test **. Significant *p* < 0.05. Read horizontally, the same numbers show differences that are not statistically significant by Tukey test. At 1/10 dilutions, read vertically, the same lower-case letters show differences that are not statistically significant according to the Student’s *t* test. At 1/100 dilutions, read vertically, the same upper-case letters show differences that are not statistically significant by the Student’s *t* test.

## Data Availability

Data associated with the article are available on the Digibug Project page at https://hdl.handle.net/10481/78885 (accessed on 11 January 2023).

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
