# Peer review of "Antimicrobial Activity and Cytotoxicity of Nonsteroidal Anti-Inflammatory Drugs against Endodontic Biofilms"

_antibiotics, 2023, doi:10.3390/antibiotics12030450_

Round 1
Reviewer 1 Report
This manuscript on antimicrobial activity and cytotoxicity of nonsteroidal anti-inflammatory drugs (sodium diclofenac, ibuprofen, and ibuprofen-arginine solutions) against polymicrobial endodontic biofilms is highly interesting and novel and touches on a highly important question. I find the experiment well organized and manuscript clearly written. I am strongly inclined to recommend this article for publication in Antibiotics. I only have one questions to the authors:
“Persistent infections have become a challenge in dentistry because of growing antibiotic resistance. Nonsteroidal anti-inflammatory drugs appear to be a therapeutic alternative to control biofilm infection.”
Why did you use saline solution in the control group and not antibiotics (e.g., amoxicillin or ciprofloxacin)?
Author Response
Reviewers' comments and Suggestions for Authors:
REPLY TO REVIEWER #1
Comments
This manuscript on antimicrobial activity and cytotoxicity of nonsteroidal anti-inflammatory drugs (sodium diclofenac, ibuprofen, and ibuprofen-arginine solutions) against polymicrobial endodontic biofilms is highly interesting and novel and touches on a highly important question. I find the experiment well organized and manuscript clearly written. I am strongly inclined to recommend this article for publication in Antibiotics. I only have one questions to the authors:
“Persistent infections have become a challenge in dentistry because of growing antibiotic resistance. Nonsteroidal anti-inflammatory drugs appear to be a therapeutic alternative to control biofilm infection.”
Why did you use saline solution in the control group and not antibiotics (e.g., amoxicillin or ciprofloxacin)?
Our response: We appreciate the reviewer's comment. Saline solution was used as the control — instead of antibiotics — because it scarcely affects microorganisms, which allows us to consider 100% cell viability and compare the antimicrobial action of the new solutions.
Author Response
REPLY TO REVIEWER #2
Reviewers' report
The manuscript by Ferrer-Luque et al. evaluates the antimicrobial activity and cytotoxicity of nonsteroidal anti-inflammatory drugs (NSAIDs) against endodontic biofilms in vitro. Specifically, the authors studied three irrigating solutions, namely sodium diclofenac (DCS), ibuprofen (IBP), and ibuprofen-arginine (IBP-arginine). The authors first assessed the antimicrobial activity of these solutions in ATP bioluminescence assays, finding that 4% IBP-arginine solution showed the highest antimicrobial effectiveness, followed by 4% DCS, and then by 4% IBP. The authors also tested the antimicrobial activity using confocal light scanning microscopy (CLSM) and live/dead staining, which showed no significant differences among the three solutions. Regarding cytotoxicity, the authors applied the above solutions to 3T3-L1 fibroblasts, and evaluated their viability percentage at different times (1 h and 24 h) and with different dilutions (1:10 and 1:100). At 1:10 dilution, the cytotoxicity results were statistically similar for all solutions and for both 1 h and 24 h assays. At 1:100 dilution, both IBP-arginine solutions showed significantly higher viability percentages than the DCS solutions. Taken together, the authors suggested that 4% IBP-arginine solution showed the most promising results in terms of antimicrobial activity and cytotoxicity.
The manuscript is generally well constructed. Below, I have a few comments and suggestions that need to be addressed.
- The manuscript currently lacks any explanation on why there seems to be a discrepancy between the ATP assay and the CLSM assay – the former showed significantly different antimicrobial activities among the three solutions while the latter did not. It would be great to at least comment on what could lead to the discrepancy and/or how each measurement is limited.
Response: We have commented on this issue in the Discussion section.
Revised text: page 6: Line 172. In this way, the viability results of both tests can be considered complementary, and discrepancies may be attributed to differences in the methodologies used. The ATP de-termination requires the disruption of the biofilms adhered to dentin for cell recovery, while CLSM evaluates the percentage of viable cells in situ, in a randomized number of dentin areas.
- I suggest adding a few lines in the footnote of Table 1 to explain: (1) different letters indicate significant differences between solutions; and (2) “SS” stands for saline solution.
Response: We have added in Table 1 the information suggested.
- I suggest adding a control experiment to Table 2, applying 100% the diluting agent to the fibroblasts. This should verify that there is not much cell death in the absence of any drugs.
Response: Data on the control groups (cells in DMEM) are included in Table 2.
- Line 95: “major” doesn’t seem to be the right word here. Maybe “substantial”?
Response: We have changed the word in the text.
- Line 140: I suggest rewriting the first sentence of this paragraph as “In the cytotoxicity assays, both IBP solutions …”
Response: We have reworded the sentence
Revised text:page 6: Line 192. “The values of cytotoxicity obtained by ibuprofen solutions…..
Reviewer 3 Report
I would like to sincerely thank the authors for their efforts in conducting the study and preparation of the manuscript. Kindly find my detailed comments and suggestions in the attached document.

Reviewer 4 Report
Overall, the proposal of the manuscript of redirecting universally used drugs for endodontic purposes is promising. However, changes must be done to the draft to consider it suitable for publication.
General Questions
· NaOCl and CHX are the conventional irrigants of endodontic practice. Why the authors did not include them as test groups comparing their antimicrobial and cytotoxicity effects to the new substances?
· How the authors decided on the concentrations of the test drugs to be used? Are there any preliminary studies to reach the optimized concentration (such as Microdilution or micro diffusion)? If so, added the previously published data on material and methods or your own preliminary data on supplemental material.
1) Introduction
· Add one section about the role of microbial biofilms on periapical periodontitis, describing the whole pathogenesis behind it and emphasizing how the immune response to it is critical to the disease development
· Describe the DCS` anti-inflammatory effect mechanism of action, such as done for IBP
· The authors cited the antimicrobial effect of ibuprofen against E. faecalis. There are any data suggesting that sodium diclofenac is effective against any classic endodontic pathogens? If the answer is yes, add it to the text.
2) Results
· Table 1: A proper description of the meaning of the lowercase letters is missing.
· Figure 1 revealed the diversity and robustness of the polymicrobial biofilm cultivated. A better description is needed, emphasizing the variety of microbial shapes found as their notable interaction
· The clsm results are poorly described, a comparative description of the confocal images should be added, comparing the antimicrobial effects of the test solutions
· Although the RLUs analyses suggested a higher antimicrobial activity induced by IBP-arginine, the Fig.2A looks to have less viable cells than Fig. 2C. A more representative image is needed.
3) Discussion
· What are the advantages that the inclusion of IBP, DCs, and IBP-arginine have in front of the currently widely used endodontic substances for disinfection? The authors should add this fact to the discussion
· There are other desirable properties that IBP, DCS, and IBP-arginine solutions should have to be considered as promising irrigants solutions. Add a section to the discussion describing them.
· (line 101-102) Include here the citations of the publications that suggest that NSAIDS have an antimicrobial and antibiofilm effect.
· (line 108-114) This part is very confusing and lacks experimental validation. Did the authors measure the pH of the solutions and of the biofilms before and after treatment? This additional data Is important to validate this statement. Clarify the bacterial metabolic pathways that may result in the increase of the pH related to the metabolism of arginine. Here, there is a suggestion for reference:
https://www.ncbi.nlm.nih.gov/pmc/articles/PMC5502959/
· Where is the conclusion?
1) Material and methods
· (lines 162-163) How the dentin specimens were obtained? The project was approved by the Ethical Committee for accessing any teeth biobank or by collecting them by individual donators? Please, clarify.
· (lines 195-196) Where is the methodology of preparation of the samples for Scanning electron microscopy?
· Please describe how many samples are for each test group in the text, not only in the tables.
· Only one image from each sample was obtained for quantification? How the randomization of the site of choice was performed?
1) Material and methods
· (lines 162-163) How the dentin specimens were obtained? The project was approved by the Ethical Committee for accessing any teeth biobank or by collecting them by individual donators? Please, clarify.
· (lines 195-196) Where is the methodology of preparation of the samples for Scanning electron microscopy?
· Please describe how many samples are for each test group in the text, not only in the tables.
· Only one image from each sample was obtained for quantification? How the randomization of the site of choice was performed?
Round 2
Reviewer 3 Report
I would like to thank the authors for their efforts in addressing the raised comments. Here are my comments on the authors' reply:
Line 61: use the term hepatic instead of liver.
The authors responded that the aim was to determine the potential use of NSAIDs in root canal treatments and not to compare with the gold standard NaOCl. The study aimed to test the potential of NSAIDs and arginine solutions for use in root canal treatment as endodontic irrigants. Therefore, it was reasonable to include a conventional endodontic irrigant like NaOCl as also dedicated by the treatment time included in this study.
Adding the text into the first paragraph of the manuscript is unlikely to serve the purpose of the study. The authors should make it clear to the reader why it is suggested to seek for novel agents for use in endodontic infections. Is it because the rise of antibiotic resistance? or limitations in the existing disinfections agents? If the purpose is to replace to the use of antibiotic dressing during regenerative endodontics, then the experimental protocol should have been changed accordingly.
Another question, why did not the authors used the same concentration NSAIDs and arginine solutions used in antibiofilm assays in the cytotoxicity evaluation?
Author Response
Comments and Suggestions for Authors (Manuscript ID: antibiotics-2185823)
I would like to thank the authors for their efforts in addressing the raised comments. Here are my comments on the authors' reply:
Line 61: use the term hepatic instead of liver.
Our response: This has been changed in the text.
The authors responded that the aim was to determine the potential use of NSAIDs in root canal treatments and not to compare with the gold standard NaOCl. The study aimed to test the potential of NSAIDs and arginine solutions for use in root canal treatment as endodontic irrigants. Therefore, it was reasonable to include a conventional endodontic irrigant like NaOCl as also dedicated by the treatment time included in this study.
Our response: We agree with the reviewer that is necessary include the gold standard NaOCl when irrigating solutions are used in clinical protocols, which will be the aim of future research in infected root canals (ex vivo). However, this study is just a preliminary in vitro investigation to explore the antibiofilm efficacy and cytotoxicity of NAID solutions and to determine their potential for use in endodontic treatments.
Adding the text into the first paragraph of the manuscript is unlikely to serve the purpose of the study. The authors should make it clear to the reader why it is suggested to seek for novel agents for use in endodontic infections. Is it because the rise of antibiotic resistance? or limitations in the existing disinfections agents? If the purpose is to replace to the use of antibiotic dressing during regenerative endodontics, then the experimental protocol should have been changed accordingly.
Our response:This concern has been clarified in lines 47-51.
Revised text, lines 47-51: ... In addition, the search for novel antibacterial agents for use in endodontic therapy is fundamentally determined by the resistance of microorganisms to antimicrobial substances. In this sense, great efforts are being directed toward drug repurposing for biofilm control in infectious processes where antibiotic performance is diminished due to the resistance generated by indiscriminate use.
Another question, why did not the authors used the same concentration NSAIDs and arginine solutions used in antibiofilm assays in the cytotoxicity evaluation?
Our response: In studies of cytotoxicity, the products are generally analyzed at dilutions because cultured cells are more vulnerable to the toxic effects of drugs than body tissues, where phagocytic cells and the lymph and blood channels help dilute and carry away the drug (Wall GL, Dowson J, Shipman C Jr. Antibacterial efficacy and cytotoxicity of three endodontic drugs. Oral Surg Oral Med Oral Pathol. 1972;33:230-41. DOI: 10.1016/0030-4220(72)90393-3.; Malheiros CF, Marques MM, Gavini G. In vitro evaluation of the cytotoxic effects of acid solutions used as canal irrigants. J Endod. 2005;31:746-8. DOI: 10.1097/01.don.0000157994.49432.67)
Reviewer 4 Report
- How the authors decided on the concentrations of the test drugs to be used? Are there any preliminary studies to reach the optimized concentration (such as Microdilution or micro diffusion)? If so, added the previously published data on material and methods or your own preliminary data on supplemental material.
Our response: This information has been added in the Material and Methods section.
Revised text, Lines 289-291. The concentrations of the tested solutions were decided based on previous results of diclofenac sodium against endodontic biofilms [21,22,25].
-Please, emphasize in the discussion that when you used IBP and IBP+Arg at 4% (based on diclofenac), you had satisfactory results for both antimicrobial and cytotoxicity compared to diclofenac.
- Add one section about the role of microbial biofilms on periapical periodontitis, describing the whole pathogenesis behind it and emphasizing how the immune response to it is critical to the disease development
Our response: Following the reviewer´s indication, we have included the following paragraph and new references in the introduction section.
Revised text: page 2: Lines 49-53 Apical periodontitis is an inflammatory disease caused mainly by bacteria orga-nized as microbial biofilms attached to the dentin of the root canal system. Infection progresses apically to reach and establish itself in periradicular tissues, where the microorganisms promote the immune response from the host and determine symptomatic or asymptomatic disease nature and periradicular tissue damage [9-11].
-This part would be more suitable at the beginning on the introduction, before line 31.
Figure 1 revealed the diversity and robustness of the polymicrobial biofilm cultivated. A better description is needed, emphasizing the variety of microbial shapes found as their notable interaction
Our response: We appreciate the reviewer's suggestion. The revised version includes a more detailed legend in Figure 1.
Revised Figure 1 legend: Representative scanning electron microscopy images showing the growth of polymicrobial biofilms in infected dentin samples: (a) x4000; (b) x13,000. Note the diversity of microbial composition (different colored arrows represent different species) grown within close proximity. Antagonistic and cooperative-synergistic microbial interactions play an essential role in ecological regulation and endodontic biofilm formation.
-I would suggest “(different colored arrows represent different species bacterial shape types, different species)
The CLSM results are poorly described, a comparative description of the confocal images should be added, comparing the antimicrobial effects of the test solutions
Our response: We appreciate the reviewer's suggestion and have included these data in Figure 2.
Revised Figure 2 legend: a) Representative images of 3 week -biofilms treated with Diclofenac (DCS), Ibuprofen (IBP), IBP-arginine, and saline solution (SS) visualized under CLSM after staining with Syto-9/PI. Syto-9 stained nucleic acid and emitted green fluorescence (considered as live cells), whereas damaged cells were stained by PI (red fluorescence for dead bacteria). b) the Log10 total biovolume in the groups was similar and showed a scarce but significant reduction with respect to the control. c) Mean (SD) percentage of green (live) cells by group (n=20 stacks/group). *The 4% DCS and 4% IBP-arginine groups gave the lowest viable cell percentages, without significant differences among them (p<0.001). **The 4% IBP group showed the highest percentage of viable cells, giving statistically significant differences from the other groups.
· Although the RLUs analyses suggested a higher antimicrobial activity induced by IBP-arginine, the Fig.2A looks to have less viable cells than Fig. 2C. A more representative image is needed.
Our response: Thank you for your observation and suggestion. Accordingly, we have selected a more representative image.
Image 2 seems to be a little unframed on the page. Adjust it before resubmitting the paper.
Where is the conclusion?
Our response In conclusion, NAID solutions could be recommendable for endodontic disinfection procedures (line 224).
This conclusion poorly describes the results found in the paper, contradicts what had been said in lines 217-218, and disregard the major importance of clinical studies to make the statement that it is recommendable in clinical practices.
Please, rewrite the conclusion focusing on the different results found for each substance and the importance to conduct an investigation of this treatment after that could be recommendable for clinical use.
1) Material and methods
· (lines 162-163) How the dentin specimens were obtained? The project was approved by the Ethical Committee for accessing any teeth biobank or by collecting them by individual donators? Please, clarify.
Our response: The study protocol (no. 1076 CEIH/2020) was approved by the Research Ethics Committee of the University of Granada, Spain; informed consent was obtained from all patients before collection of the microbiological samples or extracted teeth
Please, add this information underlined to the text.
Please describe how many samples are for each test group in the text, not only in the tables.
Our response This concern is indicated in the text and in the flowchart.
Remove the sections: methods, results, conclusions, and funding. If the authors would like to proceed with the flowchart, the writing should be more succinct.
Author Response
REPLY TO REVIEWER #4
Comments and Suggestions for Authors (Manuscript ID: antibiotics-2185823)
- How the authors decided on the concentrations of the test drugs to be used? Are there any preliminary studies to reach the optimized concentration (such as Microdilution or micro diffusion)? If so, added the previously published data on material and methods or your own preliminary data on supplemental material.
Our response: This information has been added in the Material and Methods section.
Revised text, Lines 289-291. The concentrations of the tested solutions were decided based on previous results of diclofenac sodium against endodontic biofilms [21,22,25].
-Please, emphasize in the discussion that when you used IBP and IBP+Arg at 4% (based on diclofenac), you had satisfactory results for both antimicrobial and cytotoxicity compared to diclofenac.
Our response: Following your recommendations, we have emphasized this matter in the new Conclusion drawn up.
Revised text, line 228: In conclusion, NSAIDs could be suitable drugs for the control of endodontic infection. In this work, ibuprofen mixed with arginine was the least cytotoxic and most effective anti-microbial solution against a polymicrobial root canal biofilm. Diclofenac sodium was more cytotoxic than both ibuprofen solutions and showed less antibiofilm efficacy than the IBP-arginine. The results obtained here should be confirmed in future investigations when the tested solutions might be recommended for clinical use.
- Add one section about the role of microbial biofilms on periapical periodontitis, describing the whole pathogenesis behind it and emphasizing how the immune response to it is critical to the disease development
Our response: Following the reviewer´s indication, we have included the following paragraph and new references in the Introduction section.
Revised text: page 2: Lines 49-53 Apical periodontitis is an inflammatory disease caused mainly by bacteria organized as microbial biofilms attached to the dentin of the root canal system. Infection progresses apically to reach and establish itself in periradicular tissues, where the microorganisms promote an immune response from the host and determine symptomatic or asymptomatic disease, and periradicular tissue damage [9-11].
-This part would be more suitable at the beginning on the introduction, before line 31.
Our response: Following your recommendation, the change has been made.
Figure 1 revealed the diversity and robustness of the polymicrobial biofilm cultivated. A better description is needed, emphasizing the variety of microbial shapes found as their notable interaction
Our response: We appreciate the reviewer's suggestion. The revised version includes a more detailed legend in Figure 1.
Revised Figure 1 legend: Representative scanning electron microscopy images showing the growth of polymicrobial biofilms in infected dentin samples: (a) x4000; (b) x13,000. Note the diversity of microbial composition (different colored arrows represent different species) grown within close proximity. Antagonistic and cooperative-synergistic microbial interactions play an essential role in ecological regulation and endodontic biofilm formation.
-I would suggest “(different colored arrows represent different species bacterial shape types, different species)
Our response: The suggest has been incorporated.
Revised Figure 1 legend: Representative scanning electron microscopy images showing the growth of polymicrobial biofilms in infected dentin samples: (a) x4000; (b) x13,000. Note the diversity of microbial composition (different colored arrows represent different species and bacterial shape types) grown within close proximity. Antagonistic and cooperative-synergistic microbial interactions play an essential role in ecological regulation and endodontic biofilm formation.
The CLSM results are poorly described, a comparative description of the confocal images should be added, comparing the antimicrobial effects of the test solutions
Our response: We appreciate the reviewer's suggestion and have included these data in Figure 2.
Revised Figure 2 legend: a) Representative images of 3 week biofilms treated with Diclofenac (DCS), Ibuprofen (IBP), IBP-arginine, and saline solution (SS) visualized under CLSM after staining with Syto-9/PI. Syto-9 stained nucleic acid and emitted green fluorescence (considered as live cells), whereas damaged cells were stained by PI (red fluorescence for dead bacteria). b) the Log10 total biovolume in the groups was similar and showed a scarce but significant reduction with respect to the control. c) Mean (SD) percentage of green (live) cells by group (n=20 stacks/group). *The 4% DCS and 4% IBP-arginine groups gave the lowest viable cell percentages, without significant differences among them (p<0.001). **The 4% IBP group showed the highest percentage of viable cells, giving statistically significant differences from the other groups.
- Although the RLUs analyses suggested a higher antimicrobial activity induced by IBP-arginine, the Fig.2A looks to have less viable cells than Fig. 2C. A more representative image is needed.
Our response: Thank you for your observation and suggestion. Accordingly, we have selected a more representative image.
Image 2 seems to be a little unframed on the page. Adjust it before resubmitting the paper.
Our response: The image 2 has been adjusted on the page.
Where is the conclusion?
Our response In conclusion, NAID solutions could be recommendable for endodontic disinfection procedures (line 224).
This conclusion poorly describes the results found in the paper, contradicts what had been said in lines 217-218, and disregard the major importance of clinical studies to make the statement that it is recommendable in clinical practices.
Please, rewrite the conclusion focusing on the different results found for each substance and the importance to conduct an investigation of this treatment after that could be recommendable for clinical use.
Our response: As recommended, we have rewritten the conclusion.
Revised text line 228: In conclusion, NSAIDs could be suitable drugs for the control of endodontic infection. In this work, ibuprofen mixed with arginine was the least cytotoxic and most effective anti-microbial solution against a polymicrobial root canal biofilm. Diclofenac sodium was more cytotoxic than both ibuprofen solutions and showed less antibiofilm efficacy than the IBP-arginine. The results obtained here should be confirmed in future investigations when the tested solutions might be recommended for clinical use.
1) Material and methods
- (lines 162-163) How the dentin specimens were obtained? The project was approved by the Ethical Committee for accessing any teeth biobank or by collecting them by individual donators? Please, clarify.
Our response: The study protocol (no. 1076 CEIH/2020) was approved by the Research Ethics Committee of the University of Granada, Spain; informed consent was obtained from all patients before collection of the microbiological samples or extracted teeth
Please, add this information underlined to the text.
Our response: The information has been added.
Please describe how many samples are for each test group in the text, not only in the tables.
Our response This concern is indicated in the text and in the flowchart.
Remove the sections: methods, results, conclusions, and funding. If the authors would like to proceed with the flowchart, the writing should be more succinct.
Our response: We are sorry that we cannot remove these sections, as they have been requested by another reviewer.